# Learning Chordal Markov Networks by Dynamic Programming

**Kustaa Kangas**    **Teppo Niinimäki**    **Mikko Koivisto**
Helsinki Institute for Information Technology HIIT
Department of Computer Science, University of Helsinki
{jwkangas,tzniinim,mkhkoivi}@cs.helsinki.fi

## Abstract

We present an algorithm for finding a chordal Markov network that maximizes any given decomposable scoring function. The algorithm is based on a recursive characterization of clique trees, and it runs in $\mathrm{O}(4^n)$ time for $n$ vertices. On an eight-vertex benchmark instance, our implementation turns out to be about ten million times faster than a recently proposed, constraint satisfaction based algorithm (Corander et al., NIPS 2013). Within a few hours, it is able to solve instances up to $18$ vertices, and beyond if we restrict the maximum clique size. We also study the performance of a recent integer linear programming algorithm (Bartlett and Cussens, UAI 2013). Our results suggest that, unless we bound the clique sizes, currently only the dynamic programming algorithm is guaranteed to solve instances with around $15$ or more vertices.

## 1   Introduction

Structure learning in *Markov networks*, also known as *undirected graphical models* or *Markov random fields*, has attracted considerable interest in computational statistics, machine learning, and artificial intelligence. Natural score-and-search formulations of the task have, however, proved to be computationally very challenging. For example, Srebro [1] showed that finding a maximum-likelihood *chordal* (or *triangulated* or *decomposable*) Markov network is NP-hard even for networks of treewidth at most 2, in sharp contrast to the treewidth-1 case [2]. Consequently, various approximative approaches and local search heuristics have been proposed [3, 1, 4, 5, 6, 7, 8, 9, 10, 11].

Only very recently, Corander et al. [12] published the first non-trivial algorithm that is guaranteed to find a globally optimal chordal Markov network. It is based on expressing the search space in terms of logical constraints and employing the state-of-the-art solver technology equipped with optimization capabilities. To this end, they adopt the usual *clique tree*, or *junction tree*, representation of chordal graphs, and work with a particular characterization of clique trees, namely, that for any vertex of the graph the cliques containing that vertex induce a connected subtree in the clique tree. The key idea is to rephrase this property as what they call a *balancing condition*: for any vertex, the number of cliques that contain it is one larger than the number of edges (the intersection of the adjacent cliques) that contain it. They show that with appropriate, efficient encodings of the constraints, an eight-vertex instance can be solved to the optimum in a few days of computing, which could have been impossible by a brute-force search. However, while the constraint satisfaction approach enables exploiting the powerful technology, it is currently not clear, whether it scales to larger instances.

Here, we investigate an alternative approach to find an optimal chordal Markov network. Like the work of Corander at al. [12], our algorithm stems from a particular characterization of clique trees of chordal graphs. However, our characterization is quite different, being recursive in nature. It concords the structure of common scoring functions and so yields a natural dynamic programming algorithm that grows an optimal clique tree by selecting its cliques one by one. In its basic form, the algorithm

is very inefficient. Fortunately, the fine structure of the scoring function enables us to further factorize the main dynamic programming step and so bring the time requirement down to $\mathrm{O}(4^n)$ for instances with $n$ vertices. We also show that by setting the maximum clique size, equivalently the treewidth (plus one), to $w \leq n/4$, the time requirement can be improved to $\mathrm{O}\big(3^{n-w}\binom{n}{w}w\big)$.

While our recursive characterization of clique trees and the resulting dynamic programming algorithm are new, they are similar in spirit to a recent work by Korhonen and Parviainen [13]. Their algorithm finds a bounded-treewidth Bayesian network structure that maximizes a decomposable score, running in $3^n n^{w+\mathrm{O}(1)}$ time, where $w$ is the treewidth bound. For large $w$ it thus is superexponentially slower than our algorithm. The problems solved by the two algorithms are, of course, different: the class of treewidth-$w$ Bayesian networks properly extends the class of treewidth-$w$ chordal Markov networks. There is also more recent work for finding bounded-treewidth Bayesian networks by employing constraint solvers: Berg et al. [14] solve the problem by casting into maximum satisfiability, while Parviainen et al. [15] cast into integer linear programming. For unbounded-treewidth Bayesian networks, $\mathrm{O}(2^n n^2)$-time algorithms based on dynamic programming are available [16, 17, 18]. However, none of these dynamic programming algorithms, nor their A* search based variant [19], enables adding the constraints of chordality or bounded width.

But the integer linear programming approach to finding optimal Bayesian networks, especially the recent implementation by Bartlett and Cussens [20], also enables adding the further constraints.[1] We are not aware of any reasonable worst-case bounds for the algorithm's time complexity, nor any previous applications of the algorithm to the problem of learning chordal Markov networks. As a second contribution of this paper, we report on an experimental study of the algorithm's performance, using both synthetic data and some frequently used machine learning benchmark datasets.

The remainder of this article begins by formulating the learning task as an optimization problem. Next we present our recursive characterization of clique trees and a derivation of the dynamic programming algorithm, with a rigorous complexity analysis. The experimental setting and results are reported in a dedicated section. We end with a brief discussion.

## 2   The problem of learning chordal Markov networks

We adopt the hypergraph treatment of chordal Markov networks. For a gentler presentation and proofs, see Lauritzen and Spiegelhalter [21, Sections 6 and 7], Lauritzen [22], and references therein.

Let $p$ be a positive probability function over a product of $n$ state spaces. Let $\mathcal{G}$ be an undirected graph on the vertex set $V = \{1, \ldots, n\}$, and call any *maximal* set of pairwise adjacent vertices of $\mathcal{G}$ a *clique*. Together, $\mathcal{G}$ and $p$ form a *Markov network* if $p(x_1, \ldots, x_n) = \prod_C \psi_C(x_C)$, where $C$ runs through the cliques of $\mathcal{G}$ and each $\psi_C$ is a mapping to positive reals. Here $x_C$ denotes $(x_v : v \in C)$.

The factors $\psi_C$ take a particularly simple form when the graph $\mathcal{G}$ is *chordal*, that is, when every cycle of $\mathcal{G}$ of length greater than three has a *chord*, which is an edge of $\mathcal{G}$ joining two nonconsecutive vertices of the cycle. The chordality requirement can be expressed in terms of hypergraphs. Consider first an arbitrary hypergraph on $V$, identified with a collection $\mathcal{C}$ of subsets of $V$ such that each element of $V$ belongs to some set in $\mathcal{C}$. We call $\mathcal{C}$ *reduced* if no set in $\mathcal{C}$ is a proper subset of another set in $\mathcal{C}$, and *acyclic* if, in addition, the sets in $\mathcal{C}$ admit an ordering $C_1, \ldots, C_m$ that has the *running intersection property*: for each $2 \leq j \leq m$, the intersection $S_j = C_j \cap (C_1 \cup \cdots \cup C_{j-1})$ is a subset of some $C_i$ with $i < j$. We call the sets $S_j$ the *separators*. The *multiset of separators*, denoted by $\mathcal{S}$, does not depend on the ordering and is thus unique for an acyclic hypergraph. Now, letting $\mathcal{C}$ be the set of cliques of the chordal graph $\mathcal{G}$, it is known that the hypergraph $\mathcal{C}$ is acyclic and that each factor $\psi_{C_j}(x_{C_j})$ can be specified as the ratio $p(x_{C_j})/p(x_{S_j})$ of marginal probabilities (where we define $p(x_{S_1}) = 1$). Also the converse holds: by connecting all pairs of vertices within each set of an acyclic hypergraph we obtain a chordal graph.

Given multiple observations over the product state space, the *data*, we associate with each hypergraph $\mathcal{C}$ on $V$ a *score* $s(\mathcal{C}) = \prod_{C \in \mathcal{C}} p(C) \big/ \prod_{S \in \mathcal{S}} p(S)$, where the *local score* $p(A)$ measures the probability (density) of the data projected on $A \subseteq V$, possibly extended by some structure prior or penalization term. The *structure learning problem* is to find an acyclic hypergraph $\mathcal{C}$ on $V$ that

maximizes the score $s(\mathcal{C})$. This formulation covers a Bayesian approach, in which each $p(A)$ is the marginal likelihood for the data on $A$ under a Dirichlet–multinomial model [23, 7, 12], but also the maximum-likelihood formulation, in which each $p(A)$ is the empirical probability of the data on $A$ [23, 1]. Motivated by these instantiations, we will assume that for any given $A$ the value $p(A)$ can be efficiently computed, and we treat the values as the problem input.

Our approach to the problem exploits the fact [22, Prop. 2.27] that a reduced hypergraph $\mathcal{C}$ is acyclic if and only if there is a *junction tree* $\mathcal{T}$ for $\mathcal{C}$, that is, an undirected tree on the node set $\mathcal{C}$ that has the *junction property (JP)*: for any two nodes $A$ and $B$ in $\mathcal{C}$ and any $C$ on the unique path in $\mathcal{T}$ between $A$ and $B$ we have $A \cap B \subseteq C$. Furthermore, by labeling each edge of $\mathcal{T}$ by the intersection of its endpoints, the edge labels amount to the multiset of separators of the hypergraph $\mathcal{C}$. Thus a junction tree gives the separators explicitly, which motivates us to write $s(\mathcal{T})$ for the respective score $s(\mathcal{C})$ and solve the structure learning problem by finding a junction tree $\mathcal{T}$ over $V$ that maximizes $s(\mathcal{T})$. Here and henceforth, we say that a tree is *over* a set if the union of the tree's nodes equals the set.

As our problem formulation does not explicitly refer to the underlying chordal graph and cliques, we will speak of junction trees instead of equivalent but semantically more loaded *clique trees*. From here on, a *junction tree* refers specifically to a junction tree whose node set is a reduced hypergraph.

# 3   Recursive characterization and dynamic programming

The score of a junction tree obeys a recursive factorization along subtrees (by rooting the tree at any node), given in Section 3.2 below. While this is the essential structural property of the score for our dynamic programming algorithm, it does not readily yield the needed recurrence for the *optimal* score. Indeed, we need a characterization of, not a fixed junction tree, but the entire search space of junction trees that concords the factorization of the score. We next give such a characterization before we proceed to the derivation and analysis of the dynamic programming algorithm.

## 3.1   Recursive partition trees

We characterize the set of junction trees by expressing the ways in which they can partition $V$. The idea is that when any tree of interest is rooted at some node, the subtrees amount to a partition of not only the remaining nodes in the tree (which holds trivially) but also the remaining vertices (contained in the nodes); and the subtrees also satisfy this property. See Figure 1 for an illustration.

If $\mathcal{T}$ is a tree over a set $S$, we write $\mathcal{C}(\mathcal{T})$ for its node set and $V(\mathcal{T})$ for the union of its nodes, $S$. For a family $\mathcal{R}$ of subsets of a set $S$, we say that $\mathcal{R}$ is a *partition* of $S$ and denote $\mathcal{R} \sqsubset S$ if the members of $\mathcal{R}$ are non-empty and pairwise disjoint, and their union is $S$.

**Definition 1** (**Recursive partition tree, RPT**). Let $\mathcal{T}$ be a tree over a finite set $V$, rooted at $C \in \mathcal{C}(\mathcal{T})$. Denote by $C_1, \ldots, C_k$ the children of $C$, by $\mathcal{T}_i$ the subtree rooted at $C_i$, and let $R_i = V(\mathcal{T}_i) \backslash C$. We say that $\mathcal{T}$ is a *recursive partition tree (RPT)* if it satisfies the following three conditions: (**R1**) each $\mathcal{T}_i$ is a RPT over $C_i \cup R_i$, (**R2**) $\{R_1, \ldots, R_k\} \sqsubset V \setminus C$, and (**R3**) $C \cap C_i$ is a proper subset of both $C$ and $C_i$. We denote by $\mathrm{RPT}(V, C)$ the set of all RPTs over $V$ rooted at $C$.

We now present the following theorems to establish that, when edge directions are ignored, the definitions of junction trees and recursive partition trees are equivalent.

**Theorem 1.** *A junction tree $\mathcal{T}$ is a RPT when rooted at any $C \in \mathcal{C}(\mathcal{T})$.*

**Theorem 2.** *A RPT is a junction tree (when considered undirected).*

Our proofs of these results will use the following two observations:

**Observation 3.** *A subtree of a junction tree is also a junction tree.*

**Observation 4.** *If $\mathcal{T}$ is a RPT, so is its every subtree rooted at any $C \in \mathcal{C}(\mathcal{T})$.*

*Proof of Theorem 1.* Let $\mathcal{T}$ be a junction tree over $V$ and consider an arbitrary $C \in \mathcal{C}(\mathcal{T})$. We show by induction over the number of nodes that $\mathcal{T}$ is a RPT when rooted at $C$. Let $C_i$, $\mathcal{T}_i$, and $R_i$ be defined as in Definition 1 and consider the three RPT conditions. If $C$ is the only node in $\mathcal{T}$, the conditions hold trivially. Assume they hold up to $n - 1$ nodes and consider the case $|\mathcal{C}(\mathcal{T})| = n$. We show that each condition holds.

Figure 1: An example of a chordal graph and a corresponding recursive partition. The root node $C = \{3, 4, 5\}$ (dark grey) partitions the remaining vertices into three disjoint sets $R_1 = \{0, 1, 2\}$, $R_2 = \{6\}$, and $R_3 = \{7, 8, 9\}$ (light grey), which are connected to the root node by its child nodes $C_1 = \{1, 2, 3\}$, $C_2 = \{4, 5, 6\}$, and $C_3 = \{5, 7\}$ respectively (medium grey).

(**R1**) By Observation 3 each $\mathcal{T}_i$ is a junction tree and thus, by the induction assumption, a RPT. It remains to show that $V(\mathcal{T}_i) = C_i \cup R_i$. By definition both $C_i \subseteq V(\mathcal{T}_i)$ and $R_i \subseteq V(\mathcal{T}_i)$. Thus $C_i \cup R_i \subseteq V(\mathcal{T}_i)$. Assume then that $x \in V(\mathcal{T}_i)$, i.e. $x \in C'$ for some $C' \in \mathcal{C}(\mathcal{T}_i)$. If $x \notin R_i$, then by definition $x \in C$. Since $C_i$ is on the path between $C$ and $C'$, by JP $x \in C_i$. Therefore $V(\mathcal{T}_i) \subseteq C_i \cup R_i$.

(**R2**) We show that the sets $R_i$ partition $V \setminus C$. First, each $R_i$ is non-empty since by definition of reduced hypergraph $C_i$ is non-empty and not contained in $C$. Second, $\bigcup_i R_i = \bigcup_i (V(\mathcal{T}_i) \setminus C) = (C \cup \bigcup_i V(\mathcal{T}_i)) \setminus C = \bigcup \mathcal{C}(\mathcal{T}) \setminus C = V \setminus C$. Finally, to see that $R_i$ are pairwise disjoint, assume to the contrary that $x \in R_i \cap R_j$ for distinct $R_i$ and $R_j$. This implies $x \in A \cap B$ for some $A \in \mathcal{C}(\mathcal{T}_i)$ and $B \in \mathcal{C}(\mathcal{T}_j)$. Now, by JP $x \in C$, which contradicts the definition of $R_i$.

(**R3**) Follows by the definition of reduced hypergraph. $\square$

*Proof of Theorem 2.* Assume now that $\mathcal{T}$ is a RPT over $V$. We show that $\mathcal{T}$ is a junction tree. To see that $\mathcal{T}$ has JP, consider arbitrary $A, B \in \mathcal{C}(\mathcal{T})$. We show that $A \cap B$ is a subset of every $C \in \mathcal{C}(\mathcal{T})$ on the path between $A$ and $B$.

Consider first the case that $A$ is an ancestor of $B$ and let $B = C_1, \ldots, C_m = A$ be the path that connects them. We show by induction over $m$ that $C_1 \cap C_m \subseteq C_i$ for every $i = 1, \ldots, m$. The base case $m = 1$ is trivial. Assume $m > 1$ and the claim holds up to $m - 1$. If $i = m$, the claim is trivial. Let $i < m$. Denote by $\mathcal{T}_{m-1}$ the subtree rooted at $C_{m-1}$ and let $R_{m-1} = V(\mathcal{T}_{m-1}) \setminus C_m$. Since $C_1 \subseteq V(\mathcal{T}_{m-1})$ we have that $C_1 \cap C_m = (C_1 \cap V(\mathcal{T}_{m-1})) \cap C_m = C_1 \cap (C_m \cap V(\mathcal{T}_{m-1}))$. By Observation 4 $\mathcal{T}_{m-1}$ is a RPT. Therefore, from (**R1**) it follows that $V(\mathcal{T}_{m-1}) = C_{m-1} \cup R_{m-1}$ and thus $C_m \cap V(\mathcal{T}_{m-1}) = (C_m \cap C_{m-1}) \cup (C_m \cap R_{m-1}) = C_m \cap C_{m-1}$. Plugging this above and using the induction assumption we get $C_1 \cap C_m = C_1 \cap (C_m \cap C_{m-1}) \subseteq C_1 \cap C_{m-1} \subseteq C_i$.

Consider now the case that $A$ and $B$ have a least common ancestor $C$. By Observation 4, the subtree rooted at $C$ is a RPT. Thus, by (**R1**) and (**R2**) there are disjoint $R$ and $R'$ such that $A \subseteq C \cup R$ and $B \subseteq C \cup R'$. Thus, $A \cap B \subseteq C$, and consequently $A \cap B \subseteq A \cap C$. As we proved above, $A \cap C$ is a subset of every node on the path between $A$ and $C$, and therefore $A \cap B$ is also a subset of every such node. Similarly, $A \cap B$ is a subset of every node on the path between $B$ and $C$. Combining these results, we have that $A \cap B$ is a subset of every node on the path between $A$ and $B$.

Finally, to see that $\mathcal{C}(\mathcal{T})$ is reduced, assume the opposite, that $A \subseteq B$ for distinct $A, B \in \mathcal{C}(\mathcal{T})$. Let $C$ be the node next to $A$ on the path from $A$ to $B$. By the initial assumption and JP $A \subseteq A \cap B \subseteq C$. As either $A$ or $C$ is a child of the other, this contradicts (**R3**) in the subtree rooted at the parent. $\square$

## 3.2 The main recurrence

We want to find a junction tree $\mathcal{T}$ over $V$ that maximizes the score $s(\mathcal{T})$. By Theorems 1 and 2 this is equivalent to finding a RPT $\mathcal{T}$ that maximizes $s(\mathcal{T})$. Let $\mathcal{T}$ be a RPT rooted at $C$ and denote by $C_1, \ldots, C_k$ the children of $C$ and by $\mathcal{T}_i$ the subtree rooted at $C_i$. Then, the score factorizes as follows

$$s(\mathcal{T}) = p(C) \prod_{i=1}^{k} \frac{s(\mathcal{T}_i)}{p(C \cap C_i)} . \tag{1}$$

To see this, observe that each term of $s(\mathcal{T})$ is associated with a particular node or edge (separator) of $\mathcal{T}$. Thus the product of the $s(\mathcal{T}_i)$ consists of exactly the terms of $s(\mathcal{T})$, except for the ones associated with the root $C$ of $\mathcal{T}$ and the edges between $C$ and each $C_i$.

To make use of the above factorization, we introduce suitable constraints under which an optimal tree can be constructed from subtrees that are, in turn, optimal with respect to analogous constraints (cf. Bellman's principle of optimality). Specifically, we define a function $f$ that gives the score of an optimal subtree over any subset of nodes as follows:

**Definition 2.** For $S \subset V$ and $\varnothing \neq R \subseteq V \setminus S$, let $f(S, R)$ be the score of an optimal RPT over $S \cup R$ rooted at a proper superset of $S$. That is

$$f(S, R) = \max_{\substack{S \subset C \subseteq S \cup R \\ \mathcal{T} \in \mathrm{RPT}(S \cup R, C)}} s(\mathcal{T}) \,.$$

**Corollary 5.** *The score of an optimal RPT over $V$ is given by $f(\varnothing, V)$.*

We now show that $f$ admits the following recurrence, which shall be used as the basis of our dynamic programming algorithm.

**Lemma 6.** *Let $S \subset V$ and $\varnothing \neq R \subseteq V \setminus S$. Then*

$$f(S, R) = \max_{\substack{S \subset C \subseteq S \cup R \\ \{R_1, \ldots, R_k\} \sqsubset R \setminus C \\ S_1, \ldots, S_k \subset C}} p(C) \prod_{i=1}^{k} \frac{f(S_i, R_i)}{p(S_i)} \,.$$

*Proof.* We first show inductively that the recurrence is well defined. Assume that the conditions $S \subset V$ and $\varnothing \neq R \subseteq V \setminus S$ hold. Observe that $R$ is non-empty, every set has a partition, and $C$ is selected to be non-empty. Therefore, all three maximizations are over non-empty ranges and it remains to show that the product over $i = 1, \ldots, k$ is well defined. If $|R| = 1$, then $R \setminus C = \varnothing$ and the product equals 1 by convention. Assume now that $f(S, R)$ is defined when $|R| < m$ and consider the case $|R| = m$. By construction $S_i \subset V$, $\varnothing \neq R_i \subseteq V \setminus S_i$ and $|R_i| < |R|$ for every $i = 1, \ldots, k$. Thus, by the induction assumption each $f(S_i, R_i)$ is defined and therefore the product is defined.

We now show that the recurrence indeed holds. Let the root $C$ in Definition 2 be fixed and consider the maximization over the trees $\mathcal{T}$. By Definition 1, choosing a tree $\mathcal{T} \in \mathrm{RPT}(S \cup R, C)$ is equivalent to choosing sets $R_1, \ldots, R_k$, sets $C_1, \ldots, C_k$, and trees $\mathcal{T}_1, \ldots, \mathcal{T}_k$ such that (**R0**) $R_i = V(\mathcal{T}_i) \setminus C$, (**R1**) $\mathcal{T}_i$ is a RPT over $C_i \cup R_i$ rooted at $C_i$, (**R2**) $\{R_1, \ldots, R_k\} \sqsubset (S \cup R) \setminus C$, and (**R3**) $C \cap C_i$ is a proper subset of $C$ and $C_i$.

Observe first that $(S \cup R) \setminus C = R \setminus C$ and therefore (**R2**) is equivalent to choosing sets $R_i$ such that $\{R_1, \ldots, R_k\} \sqsubset R \setminus C$.

Denote by $S_i$ the intersection $C \cap C_i$. We show that together (**R0**) and (**R1**) are equivalent to saying that $\mathcal{T}_i$ is a RPT over $S_i \cup R_i$ rooted at $C_i$. Assume first that the conditions are true. By (**R1**) it's sufficient to show that $C_i \cup R_i = S_i \cup R_i$. From (**R1**) it follows that $C_i \subseteq V(\mathcal{T}_i)$ and therefore $C_i \setminus C \subseteq V(\mathcal{T}_i) \setminus C$, which by (**R0**) implies $C_i \setminus C \subseteq R_i$. This in turn implies $C_i \cup R_i = (C_i \cap C) \cup (C_i \setminus C) \cup R_i = S_i \cup R_i$. Assume then that $\mathcal{T}_i$ is a RPT over $S_i \cup R_i$ rooted at $C_i$. Condition (**R0**) holds since $V(\mathcal{T}_i) \setminus C = (S_i \cup R_i) \setminus C = (S_i \setminus C) \cup (R_i \setminus C) = \varnothing \cup R_i = R_i$. Condition (**R1**) holds since $S_i \subseteq C_i \subseteq V(\mathcal{T}_i) = S_i \cup R_i$ and thus $S_i \cup R_i = C_i \cup R_i$.

Finally observe that (**R3**) is equivalent to first choosing $S_i \subset C$ and then $C_i \supset S_i$. By (**R1**) it must also be that $C_i \subseteq V(\mathcal{T}_i) = S_i \cup R_i$. Based on these observations, we can now write

$$f(S, R) = \max_{\substack{S \subset C \subseteq S \cup R \\ \{R_1, \ldots, R_k\} \sqsubset R \setminus C \\ S_1, \ldots, S_k \subset C \\ \forall i : S_i \subset C_i \subseteq R_i \cup S_i \\ \forall i : \mathcal{T}_i \text{ is a RPT over } S_i \cup R_i \text{ rooted at } C_i}} s(\mathcal{T}) \,.$$

Next we factorize $s(\mathcal{T})$ using the factorization (1) of the score. In addition, once a root $C$, a partition $\{R_1, \ldots, R_k\}$, and separators $\{S_1, \ldots, S_k\}$ have been fixed, then each pair $(C_i, \mathcal{T}_i)$ can be chosen independently for different $i$. Thus, the above maximization can be written as

$$\max_{\substack{S \subset C \subseteq S \cup R \\ \{R_1, \ldots, R_k\} \sqsubset R \setminus C \\ S_1, \ldots, S_k \subset C}} p(C) \prod_{i=1}^{k} \left( \frac{1}{p(S_i)} \cdot \max_{\substack{S_i \subset C_i \subseteq R_i \cup S_i \\ \mathcal{T}_i \in \mathrm{RPT}(S_i \cup R_i, C_i)}} s(\mathcal{T}_i) \right) \,.$$

By applying Definition 2 to the inner maximization the claim follows. $\square$

### 3.3 Fast evaluation

The direct evaluation of the recurrence in Lemma 6 would be very inefficient, especially since it involves maximization over all partitions of the vertex set. In order to evaluate it more efficiently, we decompose it into multiple recurrences, each of which can take advantage of dynamic programming.

Observe first that we can rewrite the recurrence as

$$f(S, R) = \max_{\substack{S \subset C \subseteq S \cup R \\ \{R_1, \ldots, R_k\} \sqsubset R \setminus C}} p(C) \prod_{i=1}^{k} h(C, R_i) , \qquad (2)$$

where

$$h(C, R) = \max_{S \subset C} f(S, R) / p(S) . \qquad (3)$$

We have simply moved the maximization over $S_i \subset C$ inside the product and written each factor using a new function $h$. Due to how the sets $C$ and $R_i$ are selected, the arguments to $h$ are always non-empty and disjoint subsets of $V$. In a similar fashion, we can further rewrite recurrence 2 as

$$f(S, R) = \max_{S \subset C \subseteq S \cup R} p(C) g(C, R \setminus C) , \qquad (4)$$

where we define

$$g(C, U) = \max_{\{R_1, \ldots, R_k\} \sqsubset U} \prod_{i=1}^{k} h(C, R_i) .$$

Again, note that $C$ and $U$ are disjoint and $C$ is non-empty. If $U = \varnothing$, then $g(C, U) = 1$. Otherwise

$$g(C, U) = \max_{\varnothing \neq R \subseteq U} h(C, R) \max_{\{R_2, \ldots, R_k\} \sqsubset U \setminus R} \prod_{i=2}^{k} h(C, R_i) = \max_{\varnothing \neq R \subseteq U} h(C, R) g(C, U \setminus R) . \qquad (5)$$

Thus, we have split the original recurrence into three simpler recurrences (4,5,3). We now obtain a straightforward dynamic programming algorithm that evaluates $f$, $g$ and $h$ using these recurrences with memoization, and then outputs the score $f(\varnothing, V)$ of an optimal RPT.

### 3.4 Time and space requirements

We measure the time requirement by the number of *basic operations*, namely comparisons and arithmetic operations, executed for pairs of real numbers. Likewise, we measure the space requirement by the maximum number of real values stored at any point during the execution of the algorithm. We consider both time and space in the more general setting where the *width* $w \leq n$ of the optimal network is restricted by selecting every node (clique) $C$ in recurrence (4) with the constraint $|C| \leq w$.

We prove the following bounds by counting, for each of the three functions, the associated subset triplets that meet the applicable disjointness, inclusion, and cardinality constraints:

**Theorem 7.** *Let $V$ be a set of size $n$ and $w \leq n$. Given the local scores of the subsets of $V$ of size at most $w$ as input, a maximum-score junction tree over $V$ of width at most $w$ can be found using $6 \sum_{i=0}^{w} \binom{n}{i} 3^{n-i}$ basic operations and having a storage for $3 \sum_{i=0}^{w} \binom{n}{i} 2^{n-i}$ real numbers.*

*Proof.* To bound the number of basic operations needed, we consider the evaluation of each the functions $f$, $g$, and $h$ using the recurrences (4,5,3). Consider first $f$. Due to memoization, the algorithm executes at most two basic operations (one comparison and one multiplication) per triplet $(S, R, C)$, with $S$ and $R$ disjoint, $S \subset C \subseteq S \cup R$, and $|C| \leq w$. Subject to these constraints, a set $C$ of size $i$ can be chosen in $\binom{n}{i}$ ways, the set $S \subset C$ in at most $2^i$ ways, and the set $R \setminus C$ in $2^{n-i}$ ways. Thus, the number of basic operations needed is at most $N_f = 2 \sum_{i=0}^{w} \binom{n}{i} 2^{n-i} 2^i = 2^{n+1} \sum_{i=0}^{w} \binom{n}{i}$. Similarly, for $h$ the algorithm executes at most two basic operations per triplet $(C, R, S)$, with now $C$ and $R$ disjoint, $|C| \leq w$, and $S \subset C$. A calculation gives the same bound as for $f$. Finally consider $g$. Now the algorithm executes at most two basic operations per triplet $(C, U, R)$, with $C$ and $U$ disjoint, $|C| \leq w$, and $\varnothing \neq R \subseteq U$. A set $C$ of size $i$ can be chosen in $\binom{n}{i}$ ways, and the remaining $n - i$ elements can be assigned into $U$ and its subset $R$ in $3^{n-i}$ ways. Thus, the number of basic operations

Figure 2: The running time of Junctor and GOBNILP as a function of the number of vertices for varying widths $w$, on sparse (top) and dense (bottom) synthetic instances with 100 ("small"), 1000 ("medium"), and 10,000 ("large") data samples. The dashed red line indicates the 4-hour timeout or memout. For GOBNILP shown is the median of the running times on 15 random instances.

needed is at most $N_g = 2 \sum_{i=0}^{w} \binom{n}{i} 3^{n-i}$. Finally, it is sufficient to observe that there is a $j$ such that $\binom{n}{i} 3^{n-i}$ is larger than $\binom{n}{i} 2^n$ when $i \leq j$, and smaller when $i > j$. Now because both terms sum up to the same value $4^n$ when $i = 0, \ldots, n$, the bound $N_g$ is always greater or equal to $N_f$.

We bound the storage requirement in a similar manner. For each function, the size of the first argument is at most $w$ and the second argument is disjoint from the first, yielding the claimed bound. ☐

*Remark* 1. For $w = n$, the bounds for the number of basic operations and storage requirement in Theorem 7 become $6 \cdot 4^n$ and $3 \cdot 3^n$, respectively. When $w \leq n/4$, the former bound can be replaced by $6w\binom{n}{w} 3^{n-w}$, since $\binom{n}{i} 3^{n-i} \leq \binom{n}{i+1} 3^{n-i-1}$ if and only if $i \leq (n-3)/4$.

*Remark* 2. Memoization requires indexing with pairs of disjoint sets. Representing sets as integers allows efficient lookups to a two-dimensional array, using $O(4^n)$ space. We can achieve $O(3^n)$ space by mapping a pair of sets $(A, B)$ to $\sum_{a=1}^{n} 3^{a-1} I_a(A, B)$ where $I_a(A, B)$ is 1 if $a \in A$, 2 if $a \in B$, and 0 otherwise. Each pair gets a unique index from 0 to $3^n - 1$ to a compact array. A naïve evaluation of the index adds an $O(n)$ factor to the running time. This can be improved to constant amortized time by updating the index incrementally while iterating over sets.

## 4   Experimental results

We have implemented the presented algorithm in a C++ program Junctor (Junction Trees Optimally Recursively).[2] In the experiments reported below, we compared the performance of Junctor and the integer linear programming based solver GOBNILP by Bartlett and Cussens [20]. While GOBNILP has been tailored for finding an optimal Bayesian network, it enables forbidding the so-called v-structures in the network and, thereby, finding an optimal chordal Markov network, provided that we use the BDeu score, as we have done, or some other special scoring function [23, 24]. We note that when forbidding v-structures, the standard score pruning rules [20, 25] are no longer valid.

We first investigated the performance on synthetic data generated from Bayesian networks of varying size and density. We generated 15 datasets for each combination of the number of vertices $n$ from 8 to 18, maximum indegree $k = 4$ (*sparse*) or $k = 8$ (*dense*), and the number of samples $m$ equaling 100, 1000, or 10,000, as follows: Along a random vertex ordering, we first drew for each vertex the number of its parents from the uniform distribution between 0 and $k$ and then the actual parents uniformly at random from its predecessors in the vertex ordering. Next, we assigned each vertex two possible states and drew the parameters of the conditional distributions from the uniform distribution. Finally, from the obtained joint distribution, we drew $m$ independent samples. The input for Junctor and

Table 1: Benchmark instances with different numbers of attributes ($n$) and samples ($m$).

| Dataset | Abbr. | $n$ | $m$ | Dataset | Abbr. | $n$ | $m$ |
|---|---|---|---|---|---|---|---|
| Tic-tac-toe | X | 10 | 958 | Voting | V | 17 | 435 |
| Poker | P | 11 | 10000 | Tumor | T | 18 | 339 |
| Bridges | B | 12 | 108 | Lymph | L | 19 | 148 |
| Flare | F | 13 | 1066 | Hypothyroid | | 22 | 3772 |
| Zoo | Z | 17 | 101 | Mushroom | | 22 | 8124 |

Figure 3: The running time of Junctor against GOBNILP on the benchmark instances with at most 19 attributes, given in Table 1. The dashed red line indicates the 4-hour timeout or memout.

GOBNILP was produced using the BDeu score with equivalent sample size 1. For both programs, we varied the maximum width parameter $w$ from 3 to 6 and, in addition, examined the case of unbounded width ($w = \infty$). Because the performance of Junctor only depends on $n$ and $w$, we ran it only once for each combination of the two. In contrast, the performance of GOBNILP is very sensitive to various characteristics of the data, and therefore we ran it for all the combinations. All runs were allowed 4 CPU hours and 32 GB of memory. The results (Figure 2) show that for large widths Junctor scales better than GOBNILP (with respect to $n$), and even for low widths Junctor is superior to GOBNILP for smaller $n$. We found GOBNILP to exhibit moderate variance: 93% of all running times (excluding timeouts) were within a factor of 5 of the respective medians shown in Figure 2, while 73% were within a factor of 2. We observe that the running time of GOBNILP may behave "discontinuously" (e.g., small datasets around 15 vertices with width 4).

We also evaluated both programs on several benchmark instances taken from the UCI repository [26]. The datasets are summarized in Table 1. Figure 3 shows the results on the instances with at most 19 attributes, for which the runs were, again, allowed 4 CPU hours and 32 GB of memory. The results are qualitatively in well agreement with the results obtained with synthetic data. For example, solving the Bridges dataset on 12 attributes with width 5, takes less than one second by Junctor but around 7 minutes by GOBNILP. For the two 22-attribute datasets we allowed both programs one week of CPU time and 128 GB of memory. Junctor was able to solve each within 33 hours for $w = 3$ and within 74 hours for $w = 4$. GOBNILP was able to solve Hypothyroid up to $w = 6$ (in 24 hours, or less for small widths), but Mushroom only up to $w = 3$. For higher widths GOBNILP ran out of time.

## 5 Concluding remarks

We have investigated the structure learning problem in chordal Markov networks. We showed that the commonly used scoring functions factorize in a way that enables a relatively efficient dynamic programming treatment. Our algorithm is the first that is guaranteed to solve moderate-size instances to the optimum within reasonable time. For example, whereas Corander et al. [12] report their algorithm took more than 3 days on an eight-variable instance, our Junctor program solves any eight-variable instance within 20 milliseconds. We also reported on the first evaluation of GOBNILP [20] for solving the problem, which highlighted the advantages of the dynamic programming approach.

**Acknowledgments**

This work was supported by the Academy of Finland, grant 276864. The authors thank Matti Järvisalo for useful discussions on constraint programming approaches to learning Markov networks.

## Footnotes

[1]We thank an anonymous reviewer of an earlier version of this work for noticing this fact, which apparently was not well known in the community, including the authors and reviewers of Corander's et al. work [12].

[2]Junctor is publicly available at www.cs.helsinki.fi/u/jwkangas/junctor/.

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
