[Reviews · NeurIPS 2014]

Submitted by Assigned_Reviewer_6

This work develops a new exact algorithm for structure learning of chordal Markov networks (MN) under decomposable score functions. The algorithm implements a dynamic programming approach by introducing recursive partition tree structures, which are junction tree equivalent structures that well suit the decomposition of the problem into smaller instances so to enable dynamic programming. The authors review the literature, prove the correctness of their algorithm and compare it against a modified version of GOBNILP, which is implements an state-of-the-art method for Bayesian network exact structure learning.

The paper is well-written, relevant for NIPS and technically sound. As far as I can tell, it is also novel and presents an important contribution to the area. The only main issue I find is the lack of a proper motivation for learning _chordal_ Markov networks. I assume the restriction to chordal networks is to allow for efficient exact inference and smaller sample complexity. The author could enlighten us with a discussion on that.

Below are some minor issues:

page 2, lines 67: There are recent publications on learning Bayesian networks of bounded treewidth which should be added as related work:

P. Parkaviainen et al. Learning Bounded Tree-width Bayesian Networks using Integer Linear Programming. In AISTATS 2014.

J. Berg et al. Learning Optimal Bounded Treewidth Bayesian Networks via Maximum Satisfiability. In AISTATS 2014.

In particular, the first publication shows an integer programming formulation of the problem which could be adapted to learn chordal MNs, similarly to the way the authors adapted the work of Barlett and Cussens (probably even easier. As the implementation of the code is freely available, the authors are suggested to compare against it.)

There is a similarity in learning maximum likelihood BNs of bounded treewidth to learning chordal maximum likelihood MNs of bounded treewidth. If that is the case, then algorithms for the former could be tested against the proposed method as well. It would also provide a better connection between these two tasks.

page 3, lines 108-110: By assuming that the scores are given as input, you implicitly assume that w (the width of the network decomposition) is small. Hence, I see little value later on in the experiments section when w is taken to infinity; the way I see it, the current approach is designed for handling cases of "small" w (perhaps up to 30 or so). A warning note here would be helpful.

same page, definition 1: it is better to give names to the conditions (1), (2) and (3), otherwise it gets confusing when following the proofs, which also contains enumerations of their own (something like RPT1, RPT2, RTP3 would do).

page 5, line 245: the enumeration (0), (1) and (3) slightly confuses the reader (which expects to see a (2) somewhere). Try renumbering the conditions.

page 7, line 370: better or additional references for score pruning rules are

C.P. de Campos, Q. Ji: Properties of Bayesian Dirichlet Scores to Learn Bayesian Network Structures. In AAAI 2010

C.P. de Campos, Q. Ji: Efficient Structure Learning of Bayesian Networks using Constraints. Journal of Machine Learning Research 12: 663-689 (2011)

pages 7-8, Experiments sec.: drawing CPTs uniformly at random seems unrealistic. It would be better to draw distributions from a symmetric Dirichlet with hyper-parameter < 1 so that it gets relatively high entropy and mimics real-world distributions. As the performance of GOBNILP is strongly affected by the parameters, using unrealistic models might bias the comparison.

page 9: write full name of JMLR in Ref. 13.
Summary: The paper is well-written, relevant for NIPS and technically sound. As far as I can tell, it is also novel and presents an important contribution on learning graphical models of bounded complexity.

Submitted by Assigned_Reviewer_17

This paper presents an algorithm to find the optimal chordal Markov network for a decomposable scoring function. In contrast to previous methods that use constraint satisfaction or linear programming, the authors show that the score of a set of junction trees factorize recursively. They then derive a dynamic programming algorithm and its complexity and show some simulation experiments.

Compared to the methods out there, this approach to structure learning of Markov networks is quite different and figuring out how to cast the problem such that it can take advantage of dynamic programming is clever. It is clear from the experiments that this method is much faster than GOBNILP. However, it's unlikely to scale given its exponential complexity in the number vertices. It's not clear to me whether the authors actually tested the constraint satisfaction approach (Corander et al) on the same instance as their method, for speed or simply made the statement that they were faster.

I would have liked the authors to organize and explain the proofs better as they were very difficult to follow. Please also label the axes in Figure 2.
Summary: The paper could be more clearly written, but otherwise a novel and interesting approach to exact structure learning of chordal Markov networks that beats the state of art significantly in speed.

Submitted by Assigned_Reviewer_32

The paper presents an algorithm for finding a chordal Markov network that maximizes
any given decomposable scoring function. The algorithm is based on a recursive
characterization of clique trees, and it runs in O(4n) time for n vertices. The algorithm is shown experimentally to outperform the current state-of-the-art solution to the problem.

The is a well-written paper that makes a progress on an important problem. The dynamic programming approach is well-described and seems to be highly appropriate for the task. The presented results are encouraging.

Summary: The is a well-written paper that makes a progress on an important problem. The dynamic programming approach is well-described and seems to be highly appropriate for the task. The presented results are encouraging.
Author Feedback
Author rebuttal: We thank all reviewers for excellent comments. We are especially grateful for the suggested references and other smaller changes and will do our best to adopt them.

To Reviewer 17:

To clarify: We only ran Junctor on instances with the same number of variables as the benchmark instances used by Corander et al. We feel this is sufficient since our algorithm uses the same time on all instances with the same number of variables. Ideally we would have liked to use the exact same data, but unfortunately neither the implementation of Corander et al. nor the exact score files they use are publicly available. We will clarify the paper to make sure there is no confusion as to how the comparison was made.

To Reviewer 6:

We agree that the motivation for learning chordal networks could be stated more explicitly. The practical advantage of chordality is primarily that it enables direct and computationally efficient parameterization of and inference with the joint distribution. Chordal or "decomposable" networks are a well studied model class and frequently appear in both theory and applications, for recent work see e.g.

G. Letac and H. Massam, Wishart Distributions for Decomposable Graphs
The Annals of Statistics, 35 (2007) 1278-1323

A. Wiesel, Y.C. Eldar, A.O. Hero, Covariance Estimation in Decomposable Gaussian Graphical Models
IEEE Trans. on Signal Processing, 58 (2010) 1482-1492

C.J. Verzilli, N. Stallard, J.C. Whittaker, Bayesian Graphical Models for Genomewide Association Studies,
The American Journal of Human Genetics, 79 (2006) 100-112

We are familiar with the very recent work on learning bounded tree-width Bayesian networks and will address it in the final version. An experimental comparison to our work may however be beyond the scope of this paper, since the two problems are different, albeit closely related.

Regarding ILP: We agree that varying the hyper parameter would be interesting and, given the time, aim to organize additional experiments. We note, however, that our results for the synthetic data are well in line with the benchmark datasets and therefore we do not expect too dramatic differences.